# Immunological Insights into Cigarette Smoking-Induced Cardiovascular Disease Risk

**DOI:** 10.3390/cells11203190

**Published:** 2022-10-11

**Authors:** Albert Dahdah, Robert M. Jaggers, Gopalkrishna Sreejit, Jillian Johnson, Babunageswararao Kanuri, Andrew J. Murphy, Prabhakara R. Nagareddy

**Affiliations:** 1Division of Cardiac Surgery, Department of Surgery, Ohio State University Wexner Medical Center, Columbus, OH 43210, USA; 2Division of Immunometabolism, Baker Heart and Diabetes Institute, Melbourne, VIC 3010, Australia

**Keywords:** smoking, cardiovascular disease, inflammation, neutrophil, immune cells

## Abstract

Smoking is one of the most prominent addictions of the modern world, and one of the leading preventable causes of death worldwide. Although the number of tobacco smokers is believed to be at a historic low, electronic cigarette use has been on a dramatic rise over the past decades. Used as a replacement for cigarette smoking, electronic cigarettes were thought to reduce the negative effects of burning tobacco. Nonetheless, the delivery of nicotine by electronic cigarettes, the most prominent component of cigarette smoke (CS) is still delivering the same negative outcomes, albeit to a lesser extent than CS. Smoking has been shown to affect both the structural and functional aspects of major organs, including the lungs and vasculature. Although the deleterious effects of smoking on these organs individually is well-known, it is likely that the adverse effects of smoking on these organs will have long-lasting effects on the cardiovascular system. In addition, smoking has been shown to play an independent role in the homeostasis of the immune system, leading to major sequela. Both the adaptive and the innate immune system have been explored regarding CS and have been demonstrated to be altered in a way that promotes inflammatory signals, leading to an increase in autoimmune diseases, inflammatory diseases, and cancer. Although the mechanism of action of CS has not been fully understood, disease pathways have been explored in both branches of the immune system. The pathophysiologically altered immune system during smoking and its correlation with cardiovascular diseases is not fully understood. Here we highlight some of the important pathological mechanisms that involve cigarette smoking and its many components on cardiovascular disease and the immune systems in order to have a better understanding of the mechanisms at play.

## 1. Introduction

Tobacco smoking, a worldwide epidemic, is a major risk for cardiovascular disease (CVD) and a leading cause of death worldwide. According to the Centers for Disease Control and Prevention (CDC), more than half a million deaths occurs annually in the US. As of 2018, a total of 13.7% of adults (15.6% of men and 12.0% of women) smoke. Moreover, more than 16 million Americans live with a smoking-related disease [1]. Parallel to the health burden, the economic burden of tobacco use is estimated to exceed 100 billion US dollars in direct and indirect healthcare costs [2]. Life expectancy for smokers is 10 years shorter than for people who do not smoke. Quitting smoking before the age of 40 can reduce the risk of dying from smoking related complications by 90%. Interestingly, and contrary to common belief, second hand smokers and smokers who lower their daily consumptions of cigarettes, although reduces their risks of lung cancer, have increased risk of developing coronary artery disease (CAD) (40–50%) compared with people smoking 20 cigarettes [3].

Cigarette smoke contains more than 7000 [1,4] chemicals, and continued exposure to tobacco smoke has been associated with a wide array of diseases ranging from auto-immune and inflammatory systems to cancer and severe reproductive complications. Tobacco smoke contains at least 250 harmful chemicals, among those at least 69 are well-established carcinogens, and have been found to be related to many cancers including lung cancer, liver cancer, colorectal cancer, prostate cancer, and breast cancer [1,4,5,6]. Alongside cancer, smoking has been shown to be a critical risk factor in the onset of autoimmune diseases [5]. Smoking provides enough triggers to facilitate the development of autoantibodies favoring citrullination [7,8] that in turn aid in the formation of anti-citrullinated peptide (ACPA) antibodies to trigger rheumatoid arthritis (RA) [9,10]. ACPAs can be detected in 80% of patients with RA and were reported to be more specific than rheumatoid factors (RF) for diagnosing RA [11,12]. One of many mechanisms that involves smoking in RA is the contribution to citrullination in RA patients. Citrullination is mediated by peptidylarginine deiminases (PADs), which is also upregulated by cigarette smoking [13,14]. Chronic exposure to CS also has a prominent effect on the immune cells, such as macrophages, neutrophils, dendritic cells, T lymphocytes, B lymphocytes as well as the production and maintenance of proinflammatory cytokines such as IL-6, IL-1β, TNF-α [15].

The list of autoimmune diseases that are related to CS is exhaustive. Due to the alteration in the immune phenotypes, the immune microenvironment, genetic and environmental factors, autoimmunity can arise leading to severe consequences and long-term damage to organs. Another autoimmune disease that is related directly to the modulation of the immune system is Type 1 diabetes (T1D). In T1D, the attack on the β-cells of the pancreas is spearheaded primarily by T cells causing their destruction and thus insufficient insulin secretion [16,17]. Multiple sclerosis (MS), another autoimmune disease that targets the demyelination of neurons in the central nervous system by immune cells, has also been correlated with CS. In this case, the mechanism of action of smoking is exerted through lung irritation, as the consumption of snuff did not increase MS risk [18,19]. In MS, smoking was also attributed to accelerated disease progression, worsened prognostics, and increased the volume of demyelinating lesions as well as cerebral atrophy in smokers [20,21,22]. In a murine model of experimental autoimmune encephalomyelitis (EAE), nicotine prevented disease exacerbation while the non-nicotine component of CS accelerated the disease by accelerating demyelination [23]. Other studies have shown that CS increases IL-6, IL-13, IL-17, IL-22, CCL2, CCL3, and CXCL10, leading to the reduction of regulatory T cells numbers, necessary for the reduction of inflammation and disease outcomes [24,25]. Cigarette smoking and autoimmune diseases have been a subject of debate for a long time. However, the effect of cigarette smoking on autoimmunity is beyond the scope of this review and is reviewed in much more detail elsewhere [26].

## 2. Smoking-Induced Pulmonary and Cardiovascular Effects

Smoking is responsible for 10% of all CVD cases [27]. The association of cigarette smoking and CVD has been addressed for over 50 years, with several direct and indirect links found between smoking and onset of CVD. Containing over 7000 different chemicals, it is hard to directly link smoking to CVD [1,4], this is becausethe plethora of chemicals with sizes ranging from atoms to small particulate matter makes it extremely laborious to pinpoint a single factor involved in CVD making the habit of smoking and CS the most complex and misunderstood risk factors for CVD. Studies conducted to this date highlighted specific CS chemicals found to be correlated with over 36 CVD subtypes. One of the first studies linking smoking to CVD established that smoking reduces flow-mediated dilatation (FMD), an early marker for vascular and endothelial dysfunction [28]. The data showed a correlation between continuous smoking and impaired FMD in a dose dependent manner with a strong association to years smoked. While the reduction of endothelium-dependent dilation was reversible after 1 year of smoking cessation [29], this represented the first insight into the effect of smoking on the vascular walls. The effect of smoking on the vasculature later expanded and was shown to induce proatherogenic alteration in the vascular walls [30].

The relationship between smoking and CVD is also highlighted through several biochemical markers of cardiac dysfunction reported to be differently affected by smoking. For instance, markers of cardiac disease positively associated with ventricular disfunction (natriuretic peptides), and heart failure progression have been reported to be increased in smokers [31]. The cardiac natriuretic peptides, atrial (ANP) and B-type (BNP) are secreted in response to various signals including myocardial stretch and volume overload. These peptides are produced with their inactive amino-terminal fragments NT-proANP and NT-proBNP [32,33]; both peptides are implicated in the regulation of electrolytes, water balance, blood pressure, and the modulation of homeostasis [34]. In the ARIC study, the correlation of total packs per year in smokers was positively associated with NT-proBNP compared to non-smokers [35]. Interestingly, this correlation was also observed in secondhand smokers [36]; however, smoking cessation had a weak, yet significant inverse association with increased NT-proBNP levels [37]. Cardiac troponin-I (cTnI), another marker commonly used as an acute myocardial infarction (AMI) indicator, was found to be depressed in smokers compared to non-smokers [38]. These findings suggest a puzzling etiological link between smoking and CVD markers requiring further exploration. Nonetheless, CS’s complete and dramatic destabilization of one’s system renders it assiduously complex to assign a direct association between the fluctuation of CVD biochemical markers and smoking. In other words, does smoke directly increase/decrease CVD markers, or other changes sustained by the affected organs, such as the lungs being responsible for initiating this shift?

The effect of CS is complicated by the diversity of cardiovascular effects that results from smoking and other tobacco products. Stroke, atrial fibrillation, coronary artery disease, aortic aneurysm, hypertension, peripheral artery disease, heart failure and sudden cardiac death are two to four times more likely to occur in smokers. Another layer of complexity is added through the advent of different types of tobacco products used including e-cigarettes. Because of their low capacity to detoxify xenobiotic products, cardiovascular tissues are more sensitive to CS and appear at lower levels of exposure compared to other diseases [39,40]. The effect of smoking on CVD disease stems from the spiraling of primary risk factors involved in sustaining cardiovascular injury. for instance, elevated blood pressure highly impacted by nicotine is a major risk factor for coronary heart disease, MI, peripheral arterial disease, stroke, kidney, and heart failure [41,42,43]. Several studies conducted in the past 40 years linked proatherogenic cellular and molecular effects to CS and the onset of CVD [44]. Smoking has been shown to alter serum lipid profiles in a proatherogenic manner [45]. Secondhand smoking significantly decreased high-density lipoproteins (HDL) [46], and increased very-low-density lipoprotein, low-density lipoprotein (LDL) and triglyceride concentrations in the serum [30]. The cessation of smoking led to an increase in HDL but total cholesterol, LDL and triglyceride levels remained unchanged [47]. In addition to modulating lipids, CS has been shown to promote a pro-oxidative environment changing the quality of the lipids [48], contributing to lipid oxidation. It is well-known that oxidized LDLs are scavenged by macrophages and cause their transformation into foam cells, which are essential in plaque formation, and the acceleration of atherosclerosis [49,50,51]. Atherosclerosis is the dominant cause of CVD, including myocardial infarction (MI), heart failure, and stroke [52]. The risk of acute coronary and cerebrovascular events, including MI, stroke and sudden death is markedly increased by smoking [14,15]. The atherogenesis is accelerated in smokers, especially in the coronary arteries, carotid and cerebral arteries, aorta, and peripheral circulation. Smoking also is correlated with angina pectoris and intermittent claudication, and vasospastic angina [16]. Smoking also causes arrhythmic events, including atrial fibrillation and sudden death. Acute myocardial infarction (AMI) among smokers is associated with a larger thrombus load with less severe atherosclerosis. The atherosclerotic formation and plaque rupture are the most common and most important drivers underlying disease processes of MI. MI occurs when the blood stops flowing properly to the heart causing muscle (myocardial) injury due to lack of oxygen supply to the tissue. The blockage occurs generally because of an obstruction due to plaque buildup in the vasculature. Tobacco cigarette smoke (TC) also increases catecholamine release and activation of the sympathetic nervous system, contributing to ischemia and arrhythmia risk. Through these mechanisms, TC smoking is associated with AMI, stent thrombosis, both atrial and ventricular arrhythmias, and sudden death.

The effect of cigarette smoking extends to the inflammatory system as well. In general, smokers have elevated serum levels of C-reactive protein (CRP). CRP is an acute phase reactant primarily produced by hepatocytes in the liver in response to cytokines (IL-1 and IL-6) [53]. Although it is an objective marker of inflammation, its elevation correlates with many traditional CVD risk factors such as smoking. CRP have not only been found to be significantly increased in smokers [54,55,56,57,58,59], but their levels were found to increased significantly with smoking intensity [58,60]. Smokers also exhibit both anti- as well as pro-inflammatory cytokines such as TNF-α, IL-1β [61,62,63,64,65,66], IL-6 [53,67,68], and CCL2 (monocyte chemoattractant protein 1 (MCP-1)) [69,70,71,72], as well as elevated WBC counts including lymphocytes, neutrophils and monocytes [73]. Additionally, CS can cause the release of free radicals, damaging endothelial cells, causing irregularities in the cell membrane architecture, and developing marked functional changes such as the lower activity of endothelial NOS and the dysregulation of local thrombotic balance [74,75]. Adhesion molecules on endothelial cells, essential elements for leucocyte recruitment, are increased by smoking [76]. These derangements lead to severe physiological stress on the vasculature by acutely decreasing coronary blood flow as well as myocardial oxygen delivery [77,78], and will be reviewed in detail later in the paper.

Aside from the cardiovascular risk, cigarette and tobacco smoke have deleterious effects on the lung remodeling, and the inflammatory response that takes place under CS’s influence. Lung and CV disease frequently coexist. It has been established that CVD is a key contributor to the morbidity and mortality associated with chronic obstructive pulmonary disease (COPD), with 30% of all COPD patients dying from cardiovascular related manifestations [79]. With CS responsible for 95% of all cases of COPD, it is therefore not surprising that smoking is considered the risk factor implicated in these diseases. COPD is the third leading cause of death worldwide, and its prevalence is still on the rise [80,81]. CS contains components such as carbon monoxide, nicotine, oxidants, fine particulate matter, and aldehydes, rendering the understanding of their involvement in COPD and lung disease a complicated task [82]. Current therapies for COPD include long-acting bronchodilators and is not sufficiently focused on the pulmonary inflammatory process that underlies the pathogenesis of the disease [83]. Continuous exposure to CS, along with other environmental and genetic risk factors, exerts deleterious and systemic effects that contribute to the development of chronic comorbid diseases that further deteriorates the quality of life in these patients. However, it is hard to determine which of the cardiovascular or pulmonary diseases is considered as the onset of deregulation of the system [84,85]. COPD is a chronic inflammatory response in the lungs, and the inflammatory burden promotes structural remodeling, damage to the small and large airways, and modification of the pulmonary endothelium, impairing the elastic recoil of the lung, and reducing lung function [86,87,88]. This increase, persistent inflammatory response, and the excessive oxidative stress generated by CS allows the spill of pro-inflammatory mediators into the circulation, and further amplifies systemic diseases such as atherosclerosis and CVD [84,89,90,91].

The occurring inflammation that happens in the lung involves several cell lines including epithelial cells, alveoli, and cells of the immune system. Airway epithelial cells act as a barrier to invading pathogens and secretes mucus that helps trap invading pathogens and inhaled particles [92], secrete antimicrobial peptides [93], as well as chemokines and cytokines that serves as an initiator to the inflammatory immune response [94,95]. Upon initiation of the inflammatory response, neutrophils are amongst the first responders to the insult; they secrete reactive oxygen species (ROS), matrix metalloproteinases, and other enzymes. These products, although beneficial for elimination of the invading dangers, are also toxic and aid in alveolar destruction [96,97]. The resulting inflammatory process also attracts macrophages and T cells that may further contribute to the pathogenesis of lung inflammation [98]. Amongst the cytokines produced during the initial phase of lung inflammation are TNF-α and IL-1β, cytokines that are detrimental not only to the initiation of the inflammatory response on site, but that also have systemic effects on the immune system [99,100]. In rodents, CS and lung inflammation were shown to be correlated and dependent on IL-1 signaling. The blocking of the inflammasome in rodents subjected to CS reduced IL-1 production and with it inflammation and neutrophilia [101]. Furthermore, consistent with animal models, human bronchoalveolar lavage (BAL) as well as serum levels of IL-1β from smokers was shown to be increased [102].

Taken together, these data suggest a detrimental aspect of cigarette smoking linking both the lung inflammation and CVD. The inflammation generated in the lungs due to cigarette smoking might be an initiator for a systemic inflammatory response generating the pro-inflammatory mediators necessary for the onset of CVD. Smoking acts as a starter by activating inflammation both systemically and locally, expands the white blood cell count, increases adhesion molecules on endothelial cells and raises systemic inflammatory cytokine secretion. Furthermore, the oxidation of the LDL proteins, together with the inflammatory stimulus, renders it a very prone environment for the onset and formation of atherosclerotic plaques (Figure 1).

## 3. Conventional vs. Electronic Cigarettes

Electronic cigarettes (EC) are battery powered devices that deliver nicotine as a heated aerosol; as such, EC are free of the typical combustion products of tobacco cigarettes. EC have been marketed since 2007 and their use has markedly increased ever since. Epidemiological studies show an increase in EC users both in Europe and the USA [11,12]. The EC marketplace is evolving rapidly; however, the debate over the long-term effects of EC are still unknown and undergoing investigation. Most of the harm that is caused by tobacco is derived from exposure to combustion-generated toxicants, while with EC the nicotine is provided without the combustion products. Most of the effect of both the EC and conventional cigarettes are related to the effects of nicotine, and the effect of nicotine on cardiovascular diseases. Nicotine binds to cholinergic receptors in the brain, and addiction to nicotine is mediated by α4β2 nicotinic acetylcholine receptors (nAChRs). The CV effect of nicotine is mediated primarily through the α3β4 nAChRs [13]. The activation of nAChRs has been shown to promote hemodynamic changes, endothelial dysfunction, insulin resistance, dyslipidemia, inflammation, and changes in the myocardium. The effect of EC is closely correlated to the design of the EC device, toxicants such as acrolein and metals, as well as particle number and size distribution, the voltage of the batteries, the composition and resistance of coils, and finally how the devices are used [64].

The cardiovascular events of EC are what is expected by the amount of nicotine delivered, as well as potential cardiovascular toxicants. ECs come with a variety of battery voltage and coil resistances, and the higher those are, the higher the temperatures that are generated and the larger the effect on aldehyde exposure and the total volume of aerosols. Currently, studies regarding EC and their effect on cardiovascular diseases are controversial. Some studies showed an acute increase in blood pressure amongst EC users [18,19], while others reported no differences in resting hear rate and blood pressure [20,21]. Other studies examined the effect of EC on arterial stiffness and myocardial function, and while some found an increase in stiffness [22], others found no effect in human cohort studies [23]. We already reviewed the effect of CS on endothelial function, in vitro studies lacked this stress response on endothelial cells generated by EC aerosol [24], however, cell death and reduced cell proliferation was enhanced with certain e-liquid vapors [25]. Endothelial cell progenitors (EPCs) have been shown to be increased with EC, another nicotine effect in the absence of endothelial injury [103]. The effect of EC on the endothelial cells is not well established, though both EC and conventional CS reduced nitric oxide bioavailability and acutely increased oxidative stress [25], which lead to an increase in oxidized LDL [104]. Another effect of EC is the activation of the “spleno-cardiac axis”. The spleno-cardiac axis is a process of inflammation that links the brain, autonomic nervous system, the bone marrow, and the spleen to the development of atherosclerosis and AMI. Recent studies in mouse models suggest that during acute stress, increased central sympathetic outflow activates bone marrow progenitor cells and leukocytes via the β-3 receptor. Leukocyte progenitors then migrate to the spleen where they proliferate, leading to an increase in the number of pro-inflammatory monocytes that then exit the spleen to the circulation, where they come into contact with the arterial wall; this increase in recruitment promotes and accelerates the progression of atherosclerosis [105].

## 4. Effect of Smoking on Inflammation and Immune System

The effect of smoking on the immune system is not well identified. Evidence from previous studies suggests that smoking compromises the immune system and increases susceptibility to infection; amongst other effects, cigarette smoking is a major risk factor for cardiovascular diseases [15,106,107]. Cigarette smoke has been implicated in the production of many immune or inflammatory mediators [64]. The immune system is comprised of both an innate and an adaptive branch, both of which are affected by cigarette smoking. Although robust and not easily deterred, the immune system under constraint and years of constant aggression is susceptible of exhaustion, dramatic change, altered function and dysregulation.

We now look at the different aspects of modulation that happen with chronic CS.

## 5. Effect of Cigarette Smoking on the Adaptive Immune System

T cells are one of the most abundant adaptive cells in our system. In general, T cells are categorized into effector T cells, helper T cells and regulatory T cells. BAL from smokers showed an increase in effector and a decrease in regulatory T cells when compared to non-smokers [105]. Most of the studies regarding adaptive immunity and smoking was done in patients with COPD. The homeostasis of circulating T cells was disrupted in many of the smokers when compared to non-smokers. Additionally, Th17 and Th1 circulating T cells were found to be elevated in smokers. In mice, COPD induction resulted in higher Th17 as well as the upregulation of related cytokines (IL-6, IL-17 and IL-23) in the lung tissue [103,108]. The CD8 population of T cells, also known as the cytotoxic T cell subset, (CTLs) plays an important role in killing infected or damaged cells and were shown to be a key mediator in the onset of COPD. Both CD8+ numbers as well as their activation were shown to be increased in many smokers [80,81,83,104]. When it comes to regulatory T cells (Tregs), whose main role is the maintenance of homeostasis and governance of the immune activation by secreting anti-inflammatory cytokines, it was shown that the percentage of Tregs in smokers was decreased, leaving the immune system in a constant state of activation [92,93]. Another type of adaptive immune cell, the B cell, which is responsible for antibody secretion, was also found to be altered in smokers. B cells are divided into different subsets, some of which have a function in innate immunity while others play important roles in the adaptive branch of the immune system. In smokers, peripheral blood memory B cells were shown to be class switch to the IgG+ phenotype [94,95]. Smokers also presented an elevated circulating immunoglobulin E (IgE) phenotype indicating a possible deleterious effect of B cells by promoting atopic diseases and asthma [96]. In summary, the effect of smoking on the adaptive immune system is shown to be dramatic and well deserving of attention. However, the initial insult is thought to be driven by the innate immune system. While the innate immune system comprises a multitude of cell types, here we focus on the effect of smoking on macrophages and neutrophils.

## 6. Effect of Cigarette Smoking on Macrophages

Generally, macrophages are polarized into activated M1 macrophages, also called “classically activated”, and the M2 macrophages, otherwise termed “alternatively activated” macrophages. The latter produces anti-inflammatory molecules such as IL-10 and TGF-b [109,110,111], while the former is responsible for a pro-inflammatory signature, secreting cytokines such as TNF-a, IL-6, IL-12, and IL-1b [109]. The renewal of macrophages depends on monocytes that are recruited to the site of injury during inflammation, and it has been suggested that the environmental cues induce polarization states [112]. In addition, distinct monocyte subsets with distinct roles during inflammation and homeostasis might give rise to different types of macrophages after recruitment in different chronic diseases such as atherosclerosis, rheumatoid arthritis, and COPD [113,114,115,116].

While there is no concluding evidence linking macrophages, cigarette smoking and CVD, we will try and enumerate the different changes brought by CS to the macrophage population, the role of macrophages in establishing and maintaining CVD, and try to understand how CS can directly or indirectly affect CVD through modification of the macrophage population.

CS is one of the many environmental stimuli that macrophages are exposed to. CS directly damages the lung and can lead to a variety of pathologies including COPD, asthma, and pulmonary fibrosis [117,118]. Although the literature regarding the effect of smoking on the macrophage population are somewhat contradictory, there is clear evidence that all smokers have significantly higher numbers of macrophages in their lungs [119]. This increase of macrophages plays a significant role in shaping the proinflammatory aspect of lung related diseases but can also be a major contributor in the dysregulation of the reparative processes that ensues [120]. The main paradigm regarding the effect of CS on macrophages is related to their phenotypic activation state. Some studies found that exposure to CS downregulates the M1 in favor of their M2 phenotype [121,122]; this is especially hallmarked by reduced pro-inflammatory cytokines [123,124,125] during infection. CS also affected macrophage viability by affecting major cytoplasmic organelles including the mitochondria, endoplasmic reticulum, and lysosomes [126]. Other studies encountered an increased M1 activity and heightened pro-inflammatory signature following CS in macrophages [127,128]. Whether or not the effect of CS on macrophage favors the pro- or anti-inflammatory response, it is clear that many factors are involved in the different studies, such as environmental factors, the existence of comorbid diseases, the type of cigarettes used, the frequency of exposure as well as the focus of study (i.e., focus on COPD or focus on infectious diseases) may all lead to different readouts regarding the effect of cigarette smoke and the macrophage population.

Alveolar macrophages (AMs) represent approximately 90% of immune cells occupying the lungs under homeostatic conditions Lugg5. As sentinels, AMs play important roles in the clearance of infectious, allergic as well as toxic particles from the alveolar-blood interface; they exert their regulatory effects via phagocytosis, the production of inflammatory mediators (ROS), cytokines (IL-1, IL-2, IL-4, IL-6 and IL-8) as well as TNF-a and IFN-g.

AMs are also professional antigen presenting cells, markedly paving the ground to other immune cells to respond appropriately to chronic and consistent insults. AMs also play a critical role in the resolution of the inflammatory processes through the secretion of anti-inflammatory cytokines such as IL-10, and after the successful clearance of a pathogen, their phagocytic capabilities extend to phagocyte neutrophils that undergo programmed cell death [129,130]. Chronic exposure to CS has many deleterious effects on AMs and recruited cells from both the innate and the adaptive immunity. The broncho-alveolar lavage of smokers shows a marked increase of immune cells infiltrating the lungs [131], with AMs representing most of the recruited portion [132]. This rapid increase in innate immune cells in the lungs is in part attributable to the CS increase in NF-kB and IL-8 in AMs and bronchial epithelial cells [133,134,135].

In alveolar macrophages, those deleterious events can include but are not restricted to: (a) the increase of alveolar macrophages in the BAL of smokers [136], with a heightened level of lysosomal enzymes and secreted elastase [137], as well as a higher myeloperoxidase activity [138], (b) smoking can also alter the phenotype as well as the metabolic response in human alveolar macrophages [139], and finally (c) alters their phagocytic abilities, especially when it comes to the resolution of inflammation through the clearing of apoptotic neutrophils [140]. The constituents of CS are distributed within two phases: the gas phase and the particulate phase [141] (Tobacco Residues (tar)). The tar is composed predominantly of electrically charged semi-liquid particles that do not pass the alveolar walls and exert local toxicity in limited organs such as the lungs, tongue, and pharynx [142]. On the other hand, components of the gas phase can pass through the pulmonary circulation to the blood stream leading to systemic effects. Alveolar macrophages are particularly sensitive to CS as both phases exert immunomodulatory effects on their functions [143]. Nicotine allows, through its receptor on macrophages (Nicotinic Acetylcholine Receptors (nAChRs)), a reduction in the clearance of respiratory bacteria *L. pneumophilia,* and inhibits the alveolar macrophage secretion of IL-6 IL-12 and TNF-α but increases IL-10 after the induction of infection [144]. Yanli Ouyang et al., showed that alveolar macrophages exposed to high concentrations of Tar altered the production of pro-inflammatory cytokines necessary for T cell and other immune cell priming such as IL-2, IL-1β, IL-6, IFN-γ and TNF-α. Ultrastructural differences have been observed in alveolar macrophages from smokers, with a clear difference in cytoplasmic inclusions [145,146]. Furthermore, endocytosis of CS by alveolar macrophages increased the auto-fluorescence of these cells, rendering hard to detect surface antigens [147]. Alveolar macrophages from smokers were shown to express a higher level of intracellular adhesion molecules (CD11a, CD54) necessary to interact and educate T cell responses [148], and a higher expression of proliferating cell markers (CD71), explaining the increase in alveolar macrophages from the BAL of smokers. However, the authors did not find any increase in phagocytic markers on macrophages from smokers, indicating an impaired phagocytic as well as major activities from alveolar macrophages (such as adhesion, bactericidal properties, ability of phagosome and lysosome to fuse) [139,149,150,151,152]. Gene expression from Heguy et al., showed in the context of alveolar macrophages during the pathogenesis of COPD that certain specific genes are altered. Most of the downregulated genes belonged to inflammatory responses, cell adhesion, the extracellular matrix, proteolysis, lysosomal function, antioxidant-related functions, signal transduction and the regulation of transcription [153].

Regardless of CS effect on macrophagic phenotype, function, and activation states, it is well established that dysregulation in the macrophage population whether it is pro-, or anti-inflammatory, has a direct effect on the cardiovascular system, and the initiation as well as progression of CVD. We have previously described how CS alter the lipid metabolism and promote a pro-oxidative environment. The deposition of oxidized-LDL and the modifications brought forward by the inflammatory response caused by smoking in the lungs further accentuate atherosclerosis that in turn contributes to CVD. In conclusion, chronic exposure to CS helps in the perturbations in the immune system that under consistent and chronic exposure alters other pathways involved in the initiation and aggravation of CVD.

## 7. Effect of Cigarette Smoking on Neutrophils

Neutrophils are amongst the arsenal of first responders to insult, they are regarded as short-lived differentiated phagocytes lacking considerable gene expression and regulatory roles in innate or adaptive immune responses. However, that classical view of neutrophils biology and function is now changing, and the existence of heterogenous populations have been suggested. Neutrophils are being proposed to be present in several organs as the resident population of inflammatory cells, able to provide inflammatory as well as anti-inflammatory responses, adapt and shift their subsets according to the environment that surrounds them [154,155,156]. Neutrophils can; (a) migrate to lymph nodes and initiate adaptive immune responses, and play the role of antigen presenting cells (APC) [157], (b) extrude their nuclear or mitochondrial DNA as neutrophil extracellular traps (NETs) through NETosis [158], (c) act as myeloid derived suppressor cells (MDSCs), and therefore suppressing the adaptive immunity and repressing cytokine production [154,159], and (d) act as pro-inflammatory mediators through the secretion of TNF-α, IL-17 and IFN-γ [159]. However, due to their short-lived cycles they are constantly renewed (granulopoiesis) and released from the bone marrow (BM) to the blood where they patrol the circulation until they are recruited by inflammation. Neutrophils are part of the first immune cells to answer infection, and the increased susceptibility to infection (*Streptococcus pneumonia*, *Neisseria meningitis*, *Haemophilus influenza*, *Pseudomonas aeruginosa*, *Mycobacterium tuberculosis*, and *Legionella pneumophilia*) [160] generated by CS in the lungs renders it an important site for immune cell influx and more specifically macrophages and neutrophils. The function of neutrophils in smokers is altered when compared to nonsmokers, which may further complicate and compromise the host defenses, and lead to serious and chronic inflammatory pulmonary as well as systemic diseases (asthma, COPD, lung cancer, RA, CVD). Neutrophils in lungs of smokers have been shown to be dramatically increased when compared to nonsmokers [161]. However, methodological studies of neutrophils from smokers showed impairment in their functions. In mice, neutrophils exposed to total particulate matter (TPM) from CS showed reduced TNF-α secretion, as well as a diminished bacterial killing activity through reduced NADPH oxidase and nitric oxide synthase (iNOS). In their study, Zhang et al., showed a neutrophilic inability to induce gp90 and iNOS expression following LPS challenge; this is due to a neutrophil reprogramming that fails to induce STAT1 activation, thus leading to a defect in bacterial killing ability [162]. The manner of cell death is a critical element in skewing the inflammatory response and its intensity. Macrophage phagocytosis of apoptotic cells has been shown to be non-phlogistic, while phagocytosis of autophagic and necrotic cells induces pro-inflammatory reactions from macrophages [163]. Neutrophils exposed to cigarette smoke extract (CSE) triggered an atypical cell death that includes features of apoptosis, autophagy, and necrosis. Furthermore, atypically dying neutrophil exhibit “eat me” surface markers; however, they elicit no inflammatory response from macrophages [164]. Other studies have shown that CSE have a direct effect on neutrophils, this includes their increased numbers in the circulation, increased expression of nicotinic and formyl-methionyl-leycyl-phenylalanine receptors, as well as an impairment of chemotaxis and phagocytosis [165,166]. Neutrophils are known for their secretion of ROS. In the context of cigarette smoking, the data has been controversial; while some advance that smoking attenuates the respiratory burst of neutrophils [167], others found no effect [168], and even an increased effect on respiratory burst of neutrophils under cigarette exposure [169].

Neutrophil extracellular traps (NETs) are formed by activated neutrophils that undergo a unique form of cell death known as NETosis [170]. NETs are composed of chromatin containing specific proteins from neutrophilic granules (elastase) as well as histones [171]. NETs are necessary armaments of neutrophils for confining and eliminating invading microbes. However, an extensive NET formation exerts a deleterious effect in the host and can be the onset of a variety of autoimmune diseases [172]. Substantial controversy exists pertaining the effect of CS and its effect on NET formation. White et al. showed that CSE had a direct impact on NET formation by inhibiting the PKC signaling pathway; however, CSE was not the immediate effect on HOCl net formation, as it is activated downstream of PKC. Moreover, neutrophils exposed to CSE showed reduced speed, velocity, and directional chemotaxis in response to fMLP and IL-8, indicating that the CS extract presented inhibitory effects on the chemotaxis of neutrophils [173]. On the other hand, chronic exposure to CSE was shown to increase NET release. In their study, Qiu et al. demonstrated that neutrophils from chronic CS-exposed mice were more prone for NET deployment by CSE in vitro. Furthermore, they showed that NET formation in mice exposed to chronic CS was not cleared efficiently in the serum of smoking mice, indicating a prolonged effect of NETs on triggering the immune response and more specifically plasmacytoid dendritic cells (pDCs). The activation of these pDCs by prolonged NET exposure favored pDC secretion of IFN-γ, IL-12 and IL-6 and favored the skewing of T cells towards the Th1 and Th17 phenotypes that further fueled the ongoing inflammatory response [174]. The production of IL-1β from cigarette smoking indicates an activation of the inflammasome. Wu et al. showed that gene expression of Nlrp3, caspase 1 as well as proinflammatory levels of IL-1β and IL-18 were increased in endothelial cells of ApoE smoking mice vs. control mice; furthermore, ApoE mice subjected to nicotine showed larger atherosclerotic lesions compared to control mice. Furthermore, cells that were exposed to nicotine showed the high expression of Caspase 1 and NLRP3, an effect that was also measured by the levels of IL-1β and IL-18 production. In addition, Wu et al. linked inflammasome activation to ROS levels, as the blocking of ROS production completely inhibited the inflammasome and pyroptosis cell death of endothelial cells [175].

The dramatic increase of neutrophils is not solely attributed to CS; high neutrophil counts in 12 types of CVD, encompassing more than 7000 patients, showed a higher incidence in CVD [176,177,178,179,180,181]. Another study by Shah et al. found a direct correlation between high neutrophil numbers and hypertension [179], recurrent ischemic events, and mortality in CAD [176,177], left atrial thrombosis [181], and impaired myocardial perfusion [180]. In addition to their increased numbers, the neutrophil activation state is an important factor in determining and developing CVD. For instance, during hyperglycemic conditions, activated neutrophils secrete large quantities of the S100 calcium binding proteins A8/A9 (S100A8/A9) that in turn stimulate hepatic Kupffer cells resulting in IL-6 production and secretion. IL-6 acts on hepatocytes, stimulating thrombopoietin production causing inflammatory platelet hyperplasia [182]. IL-6 is a pleotropic cytokine that plays a central role in mediating inflammation and is the central stimulus for acute-phase response [183]. High plasma concentrations of IL-6 have been associated with the risk of myocardial injury, heart failure and mortality [184], and it is also responsible for the induction of hepatic synthesis of C-reactive protein (CRP), a known proinflammatory maker that has also been associated with cardiovascular events [67,185]. Hyperglycemia allegedly increases the risk of CVD. Hyperglycemia not only promotes the generation of primed neutrophils [53] but has been shown to also promote NET formation through NADPH oxidase and ROS production [57,58]. It has also been shown that hyperglycemia activates renin-angiotensin II. Angiotensin II is a vasoconstrictor that plays an important role in hypertension and atherosclerosis and the production of ROS in neutrophils which leads to NET formation [61]. NETs play a central role in thrombosis by promoting fibrin deposition and the formation of fibrin networks [186], and express the functional tissue factor that induces platelet activation and thrombin generation [187]. Nucleosomes (DNA-histone complexes) and double-stranded DNA (dsDNA) biomarkers of net formation were identified in increased numbers and correlated with a series of CV events in patients [188,189,190,191]. In addition to the role of NETs, neutrophils are major contributors to the establishment and acceleration of atherosclerosis, but also promote atherosclerotic plaque instability [192,193,194,195,196,197], which is very well documented in another review [198].

All throughout this review we have highlighted changes brought by CS on the system, where the dysregulation of multiple facets together is able to shift and influence the immune system. Metabolic changes and inflammatory imbalance together can cultivate an environment that favorizes CVD. In other words, the perturbances brought by CS are able to modify the environment to favor CVD, both from the metabolic as well as the immune point of view. Finally, trying to understand the role of CS in CVD, one has to keep in mind the development of CVD is at its core related to fundamental changes brought by the accumulation of chronic metabolism and immune defects, defects that are highly induced by CS.

This data provided evidence of the physiopathology of nicotine, cigarette smoking and consumption of other tobacco related products as a pro-atherosclerotic inducer. Taken together, these studies have shown an important role for neutrophils during chronic exposure to CS; although some hypotheses are controversial, further studies are needed to establish the true nature of neutrophils’ implication in smoking.

## 8. Conclusions

Ample evidence from decades of research has led to the identification of a wide array of CVD risk factors and biomarkers that are increased during tobacco consumption. The emergence on the market of new tobacco products, the extensive multitude of chemical components generated by cigarette combustion, the years of continued abuse exerted by cigarette smoke, and the lack of CVD biomarkers specific to tobacco products renders it challenging to pinpoint the exact effect of damage caused by cigarette smoke on the cardiovascular system. The changes brought by CS on the different risk factors (BP, dyslipidemia, and insulin resistance) that contribute to CVD makes it unclear to what extent each of the risk factors is involved in the overall morbidity and mortality related to CVD. The evidence provided in this review aimed at exposing the deleterious effect of cigarette smoking on the immune, inflammatory, pulmonary, and vascular systems, interrupting general homeostasis and leading to deleterious effects that cumulates with chronic and continuous exposure to cigarette smoking, but also helped in positioning the current ineffectiveness and non-specific state of biomarkers targetting CS and CVD. In this regard, the measurements of biomarkers require to evolve and adapt to the changing aspects of tobacco products, behavioral habits of consumption, different aspects of risk factors involved in CVD as well as risk factors that emerge from the different deregulation of the various systems involved in CS.

Although these challenges make it seem as though smoking has no specific effect on CVD due to the vague and somewhat indirect correlation, this is, an erroneous assumption, as the collective accumulated data provide enough overwhelming and convincing evidence to support the significant adverse cardiovascular effects that are brought on by tobacco products. The evidence discussed in this review is sufficient to promote smoking cessation. It takes almost a year following smoking cessation for the CVD risk to decrease by about 50%, and another 15 years to return to non-smoker levels. However, marked health improvements are observed as early as 24–72 h following cessation, and dramatic improvements are observed, with carbon monoxide levels decreasing and oxygen levels returning to normal, a recovery of the sense of smell and taste, and breathing becomes easier. With health benefits improving and accruing with time, tobacco abstinence is the best course of action for smokers to improve their overall well-being and ensure their cardiovascular health.

We have previously demonstrated the impact of deregulation of the innate immune system in shaping the progression of CVD, more specifically MI. We propose that in the context of CS, the alteration of epithelial cells and the inflammatory environment that ensues in the lungs allows innate immune cells such as neutrophils to become activated by damage associated molecular patterns (DAMPs) generated by the inflammatory microenvironment in the lungs. This will in turn lead to their activation, further recruitment, as well as attracting of other cells of the immune system to the area. Once activated, neutrophils are known to become hyper-activated, generate high amounts of ROS, and release a variety of proteolytic enzymes that could exacerbate local and systemic tissue injury. With nicotine and other CS constituents already causing damage to the vasculature, lungs, and altering the activation of the inflammatory pathways, we believe that, added together, these phenomena will entice the onset and the accelerated development of cardiovascular diseases. We are still far from fully depicting and understanding the complex interplay between the different systems and their contribution towards CVD.

## Figures and Tables

**Figure 1 cells-11-03190-f001:**
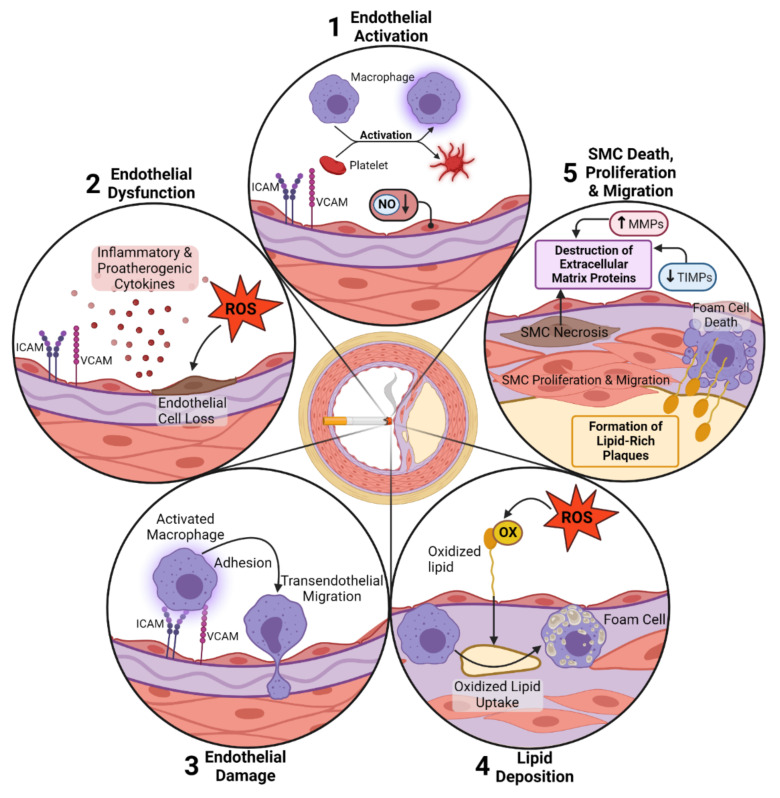
Schematic representation of smoking-induced signaling pathways in the vessel wall. (**1**) Cigarette smoke–induced oxidative (OX) stress was shown to activate the endothelium by induction of adhesion molecule expression (e.g., intracellular adhesion molecule, vascular cell adhesion molecule), as well as macrophages and platelets. Endothelial activation is characterized by the reduction of NO levels within the cells and resulting in the loss of function of smooth muscle cells (SMCs) in the vessel media. (**2**) In response to smoke exposure, endothelial cells are known to release inflammatory and proatherogenic cytokines. All of these processes lead to endothelial dysfunction. The direct physical effects of smoke compounds and produced ROS lead to endothelial cell loss by apoptosis or necrosis. (**3**) Besides endothelial cells, macrophages are activated by the expression of adhesion molecule receptors recognizing adhesion molecules on endothelial cells. (**4**) After adhesion and trans endothelial migration, macrophages take up oxidized lipids produced by oxidative modification through smoke-increased ROS production. The scavenger receptor-mediated uptake of lipids induces the formation of so-called foam cells within the aortic wall, and the subsequent death of foam cells induces the release of these lipids and the formation of lipid-rich aortic plaques. Similarly, it is postulated that smoking induces an increase in SMC proliferation and migration provoking intimal thickening and plaque formation. (**5**) The triggering of SMC death by necrosis is a further consequence of exposure to smoke that triggers inflammatory signals, as well as the release of intracellular proteolytic enzymes inducing the cleavage of extracellular matrix proteins. The destruction of extracellular matrix proteins is further enhanced by the increased expression of matrix metalloproteinases (MMPs) and the reduced expression of tissue inhibitors of MMPs (TIMPs).

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
