# Peer review of "Immunological Insights into Cigarette Smoking-Induced Cardiovascular Disease Risk"

_cells, 2022, doi:10.3390/cells11203190_

Round 1
Reviewer 1 Report
In the paper, " Immunological Insights into Cigarette Smoking-Induced Cardiovascular Disease Risk " Dr. Albert Dahdah and colleagues aims to synthesize novel conceptual advances on the immunomodulatory action of cigarette smoke (CS) with a focus on the cardiovascular disease risk. Targeting CS induced immune disorder signaling is proposed as a potential preventive and therapeutic strategy in cardiovascular disease (CVD).
Major comments: a major concern is that the author often focuses on “Immunological Insights into Cigarette Smoking-Induced Cardiovascular Disease Risk” but most reviewed studies do not have a direct link to cardiovascular pathological changes except of “Smoking-Induced Pulmonary and Cardiovascular Effects” Line 74 – 111.
- The author did not review how CS associated lung injury affected cardiovascular disease, Line 112 – 135. e.g. smoking related pulmonary hypertension and heart disease.
- The author only reviewed one group study about endothelial cell dysfunction and atherosclerotic pathogenesis (Line 329), but did not mention how macrophage, neutrophil and immune dysfunction cause cardiovascular lesions under smoking condition, Line 201 -329.
3. Figure 1: It need to add more studies related to Figure 1. smoking-induced signaling pathway in the vessel wall. e.g. smoke and adhesion molecules ICAM, VACM; smoke and endothelial; smoke and SMC.
In addition, the paragraphs would be easier to read if numbers are added to the figure and figure legend. e.g.
Figure 1. Schematic representation of smoking-induced signaling pathways in the vessel wall. (1). Cigarette smoke–induced oxidative (OX) stress was shown to activate the endothelium by induction of adhesion molecule expression (e.g., intracellular adhesion molecule, vascular cell adhesion molecule), as well as macrophages and platelets. Endothelial activation is characterized by the reduction of NO levels within the cells and resulting in loss of function of smooth muscle cells (SMCs) in the vessel media. (2). In response to smoke exposure, endothelial cells are known to release inflammatory and proatherogenic cytokines. All these processes lead to endothelial dysfunction. Direct physical effects of smoke compounds and produced reactive oxygen species (ROS) lead to endothelial cell loss by apoptosis or necrosis. (3). Besides endothelial cells, also macrophages are activated by the expression of adhesion molecule receptors recognizing adhesion molecules on endothelial cells. (4). After adhesion and trans endothelial migration, macrophages take up oxidized lipids produced by oxidative modification through smoke-increased ROS production. Scavenger receptor-mediated uptake of lipids induces the formation of so-called foam cells within the aortic wall, and subsequent death of foam cells induces the release of these lipids and the formation of lipid-rich aortic plaques. Likewise, (5). it is postulated that smoking induces an increase in SMC proliferation and migration provoking intimal thickening and plaque formation. Triggering of SMC death by necrosis is a further consequence of exposure to smoke that triggers inflammatory signals, as well as the release of intracellular proteolytic enzymes inducing cleavage of extracellular matrix proteins. Destruction of extracellular matrix proteins is further enhanced by increased expression of matrix metalloproteinases (MMPs) and reduced expression of tissue inhibitors of MMPs (TIMPs).
Minor comments
1. Line 38 - 40: if author cannot present more specific supporting evidence or information, this part should be removed from review.
Interestingly, and contrary to common belief, smoking one cigarette daily can have severe effects of developing coronary artery disease (CAD) by 40-50% compared with people who smoke 20 cigarettes.
2. The abbreviations need to be explained at first use in the text, and when once introduced, the abbreviation should be exclusively used throughout the whole text.
e. g.
Line 48 rheumatoid arthritis (RA)
Line 99 and 103 MI, myocardial infarction;
Line 308 and 334, reactive oxygen species (ROS)
Line 134, 212 and 248, bronchoalveolar lavage (BAL)
Line 95, NOS
3. It needs to verify the chemical number of cigarette smoke , 7000 in Line 41 and 4500 in Line 76, and add relative reference.
4. incomplete sentence : Line 110, Both atrial and ventricular arrhythmias, and sudden death.
5. Repeat explanation of “Tar”
Line 253, and tar, two components of cigarette smoke.
Line 256-257, Tar (another cigarette smoke component)
6. It needs the references for the author’s studies: Line 347, Our laboratory has been a pioneer in showing the cardiovascular 347 effects of altered neutrophils in the context of cardiometabolic diseases.

Author Response
Please see our responses to reviewer comments highlighted in red.

Reviewer 2 Report
The current review titled “Immunological Insights into Cigarette Smoking-Induced Cardiovascular Disease Risk”, highlights the impact of cigarette smoking on the immune, cardio, pulmonary and vascular systems. The authors discussed the molecular mechanisms of how Smoking effects both the structure and function of major organs (lungs and cardiovascular system). They have highlighted that smoking plays an important role in maintaining the homeostasis of the immune system any alterations in these pathways elicits inflammatory responses leading to autoimmune, inflammatory diseases and cancer. Most of the content discussed are not very novel. Several important points should be addressed before considering the article for publication.
Major comments –
1. Why did not the authors talk about the biochemical markers associated with cardiac injury?
2. The authors discussed how CS effects on the innate immune system in a general way. It would be better to highlight the biomarkers of cardiovascular diseases (such as C-Reactive peptide (CRP), IL-6, ICAM-1, and CCL-2).
3. Why did not the authors mention about the current challenges and future directions in association with cardiovascular disease risk caused by using Cigarette smoke.
4. As the authors highlighted about the effect of CS mainly on pulmonary, CVD and innate immune system. It would be better to make separate tabular discussing about the impact of CS on each organ. In the tabular column discus about the pathological, physiological, and metabolic effects of CS.
Author Response

(The authors gave the same response as above.)

Round 2
Reviewer 1 Report
Cells-1903471 – Review
Thank authors for providing response and revised manuscript.
Major comments
For last major comments 2, “The author only reviewed one group study about endothelial cell dysfunction and atherosclerotic pathogenesis (Line 329), but did not mention how macrophage, neutrophil and immune dysfunction cause cardiovascular lesions under smoking condition, Line 201 - 329.”
We have tried to find in the literature any plausible study investigating the intersection between smoking, the immune system, and cardiovascular diseases. Apart from the major focus on the atherosclerotic pathogenesis and the endothelial cell dysfunction little has been done on how the immune system is altered specifically regarding CVD. However, the study of the inflammatory response and its deregulation during lung injury is very well explored, especially when the lung is the primary organ affected during smoking. If the reviewer has any leads that can be helpful to improve this aspect of the review, we would be grateful.
Thank you to the authors for mentioning this. While neutrophils within the atherosclerotic lesion and smoke is less known, there are a number of different macrophage and Immune System manuscripts related to CVD. Please try to use different key words to search online ( e.g. Google and Pubmed). Below are several example papers (key searching words: smoke/cigarette, neutrophil/macrophage/ Immune System/IgE/IgA , atherosclerosis/cardiovascular) for your consideration to incorporated within the appropriate sections.
PMID: 12135614
PMID: 15050297
PMID: 36009397
PMID: 33189199
PMCID: PMC7670266
PMID: 33182063
PMID: 32245320
PMID: 32173266
PMID: 36071890
PMID: 16254210
PMID: 33626512
PMID: 34475225
PMID: 32772350
PMID: 16305810
PMID: 33877667
PMID: 34187206
PMID: 7517402
PMCID: PMC3881579
PMID: 24453429
PMID: 17210084
PMID: 30026091
Minor Comments
Line 62: remove parentheses
Line 110: add coma between troponin I (cTnI) and another
Line 392: remove coma after “Wu et al.”

Author Response
We thank the reviewer for his diligence and help regarding the comment. We have looked at the different articles as well as other articles suggested by the reviewer and made several changes in the manuscript. We want to first bring the attention to the role of neutrophil in atherosclerosis, the role of neutrophils and their NETs are known and have been recently the center of many publications, we have tried to highlight the effects without delving into details as that would be another review. It is true that the role of CS on different facets of macrophages, especially alveolar macrophages are very well known, and while their implication in CVD is very well described as well, especially in atherosclerosis, there is no direct link between the different abnormalities in macrophages due to CS and CVD. However, we did mention some cases where dysregulation of the macrophage population due to CS can be involved in the onset and progression of CVD. We hope to have answered to the best of the reviewer’s comments.

Round 3
Reviewer 1 Report
Thanks authors for their responses.